# Prompt Injection: Parameterization of Fixed Inputs

## Abstract

Recent works have shown that attaching prompts to the input is effective at conditioning Language Models (LM) to perform specific tasks. However, prompts are always included in the input text during inference, thus incurring substantial computational and memory overhead. Also, there is currently no straightforward method of utilizing prompts that are longer than the maximum input length of the LMs without incurring additional costs during inference. We propose Prompt Injection (PI), a novel formulation of injecting the prompt into the parameters of an LM to be an efficient alternative to attaching fixed prompts to the input. We show that in scenarios with long fixed prompts, PI can be up to 280 times more efficient in terms of total FLOPs than previous approaches. We further explore methodologies for PI and show promising results in persona-dependent conversation, semantic parsing, and zero-shot learning with task instructions. Through these explorations, we show that PI can be a promising direction for conditioning language models, especially in scenarios with long and fixed prompts[1].

## 1   Introduction

Contemporary works with large Language Models (LMs) [3, 32, 23, 5, 28] have shown that attaching prompts (also referred to as *prefixes*) to the input is effective at conditioning LMs to perform specific tasks. During training, LMs are trained to condition on the given prompts in hopes of generalizing to unseen prompts during inference. Unseen prompts can be a persona for persona-dependent conversation [39], database schema for semantic parsing [10], and task instruction for zero-shot learning with task instructions [23]. In these tasks, a new prompt is fixed to the input at every inference. For instance, in persona-dependent conversation [39, 18, 33], a persona description is appended to the dialogue history, so that the LM can always be conditioned on the persona. For another example, in semantic parsing, the LM is conditioned on the database schema as well as natural language questions to generalize to a new database [37, 10, 36]. Lastly, zero-shot learning with task instructions [32, 23] involves adding natural language instructions to the inputs for adapting LMs to novel tasks.

However, concatenating prompts to input sequences for prompt-dependent inference has two major limitations. (1) During inference, prompts are always included in the input text and thus incur computational and memory overhead [16]. (2) It is challenging to fit a long text such as the detailed description of a persona as a prompt into Transformer-based models whose input lengths are often fixed [27]. For instance, in persona-dependent conversation, the model constantly refers to the persona description along with the dialogue history [35, 22], as shown in the left side of Figure 1. Moreover, in real world scenarios, a persona may consist of a long detailed text description of a character or person, not just a few profile sentences. Naively concatenating long prompts to the input sequences is challenging due to the quadratic cost in time and memory of Transformer-based architectures with

---

[1]Code used for the experiments and a demo are available at this link

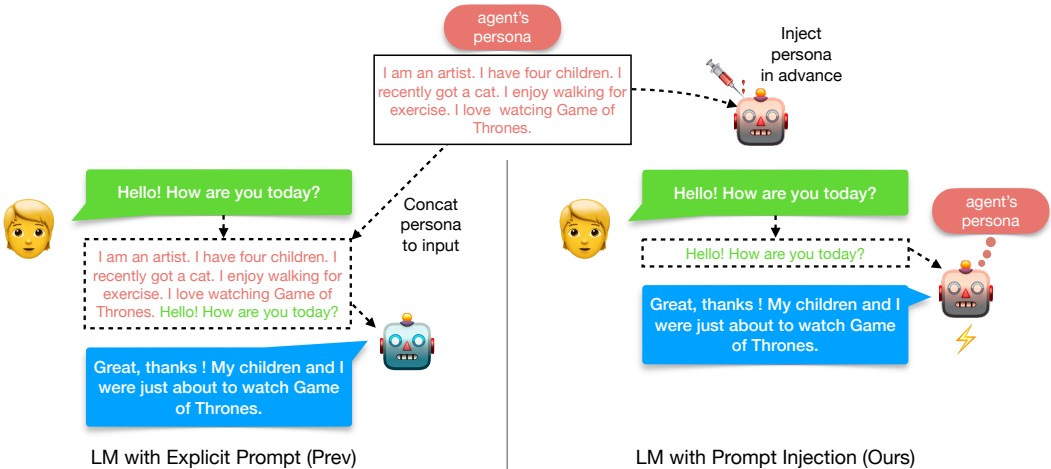

Figure 1: Prompt Injection example on a persona-dependent conversation. The left side presents an inference procedure of a previous approach where the persona (prompt) is concatenated to every input. The right side describes Prompt Injection, where the persona is *injected* into the model in advance, so that the model is able to generate responses without constantly referring to the persona description. Thus, Prompt Injection approach takes less time to generate responses than the previous method.

regard to the input sequence length. Other approaches specialized for long inputs [1, 13], such as Fusion-in-Decoder [12], or those that augment the LM with a retrieval mechanism [9] may be used but still come with increased overall memory and computations, ultimately leading to a delay in generating responses. This problem becomes critical in situations where the LMs are deployed, and fast inference speed is required.

In this work, we formulate a novel problem called Prompt Injection (PI), where we attempt to *inject* a given prompt into the parameters of an LM to address the two limitations mentioned above. With PI, LMs can produce prompt-dependent outputs without the computational overhead of appending fixed prompts at inference time (the right side of Figure 1), and it also enables the injection of longer prompts in a wholistic way. More specifically, we first show that PI is much more efficient (up to 280 times) in terms of total FLOPs compared to previous approaches that may be used for handling long prompts such as Fusion-in-Decoder [12] or Linear Transformer [13]. Next, we explore different methodologies as baselines for PI, including the continued pre-training approach on the prompt as well as a novel distillation approach called Pseudo-INput Generation (PING), in order to analyze what components are effective for successful PI. We apply these PI methods to three different tasks with fixed prompts: persona-dependent conversation, semantic parsing, and zero-shot learning with instructions. We compare the methods against LMs with explicit prompts as the upper bound (i.e., unconstrained) as well as the LM without both the prompt and PI as the lower bound. Experimental results show meaningful improvements with respect to the lower bound, but also exhibit a non-trivial gap with the upper bound. Despite the performance gap, we still believe that PI is a direction worth exploring considering the computational benefit of the injection, especially since inference speed is critical in real world applications.

In sum, our main contributions are three folds:

- We formally define the Prompt Injection (PI) formulation and demonstrate its necessity in terms of computation and memory efficiency, especially in scenarios with long prompts.

- We explore baseline approaches for PI, showing that performance can approach the upper bound (unconstrained) performance in some cases.

- We show that the *injection* of long prompts (e.g., detailed description of persona) can be achieved through PI and show its efficiency in comparison with previous methods, being up to 280 times more efficient during inference.

Through this work, we hope the community explores PI as an efficient alternative for performing prompt-dependent tasks.

## 2   Related Work

**Prompting**   Prompting is an emerging paradigm for modeling LMs, especially for few-shot and zero-shot learning [20, 3, 21, 24, 32, 23]. With the help of appropriate prompts, one can exploit knowledge learned by a pre-trained LM and manipulate the LM's behavior. The benefit of prompting is that the pre-trained LM can adapt to new scenarios with few or no labeled training data. However, for the in-context learning scenario, processing prompts that involve many training examples for each inference incurs substantial computational and memory overhead [16]. Given training data, Liu et al. [16] replace in-context learning with fine-tuning a small set of parameters for tackling the above issue. Prompt Injection also tackles the same issue but assumes a stricter scenario where there are no training data for the given prompt.

**Efficient Transformers for Long Inputs**   One can consider using efficient Transformer-based [29] architectures for handling long input sequences [27]. The main challenge of using a vanilla Transformer architecture is the quadratic cost in time and memory with regard to the input sequence length due to the self-attention operation. There has been a surge of recent works addressing this problem [6, 38, 1, 13, 40, 8]. They are primarily dedicated to improving either the efficiency of the self-attention mechanism or the general efficiency of the Transformer architecture through sparse models. Our Prompt Injection approach tackles the efficiency problem of performing prompt-dependent tasks by keeping the input sequences short (without prompts), bounding the time and memory complexity to a constant invariant of the length of the prompt.

**Persona-dependent Conversation**   Endowing a chabot with a persona [39, 18, 33] is challenging, but it enables the chatbot to deliver more personal, specific, consistent, and engaging conversations [39] and gain user trust [17, 25, 19]. To achieve this, previous works have attached a persona to the dialog history at every inference time, so that the model can always be conditioned on the persona. However, given a long persona description, this approach brings the critical problem of increased overall memory and computations, resulting in delayed response generation. An LM augmented with a retrieval mechanism [9] may be used but still comes with non-trivial computational overhead. Prompt Injection allows a dialogue agent to generate responses without a persona description as the explicit input once the persona is injected.

**Semantic Parsing**   Semantic parsing is the task of mapping a natural language query into a SQL query executable on a database. Recently, the community has focused more on cross-domain (cross-database) semantic parsing, where models are trained and tested on different domains (databases) [37]. The domain-adaptation setup introduces many generalization challenges, such as non-explicit column names and domain-specific phrases [10], and recent works concatenate the natural language query with the serialized database schema as the input to address the problem [26, 7, 36]. With Prompt Injection, the model is adapted to a new database schema in advance, so that it can map natural language queries to SQL queries on the new database without explicitly referring to the schema during inference.

**Zero-shot Learning with Task Instructions**   Recent works [23, 32] have addressed zero-shot generalization to new tasks [3, 14] by multi-task prompted training. With multi-task prompted training, the models learn to use task instructions as prompts to generalize to unseen tasks. It is demonstrated that this approach improves generalization ability to novel tasks and offers an effective substitute for unsupervised language model pre-training. Through Prompt Injection, the LM can be aware of a novel task instruction before performing the task and thus does not require the instruction, which can be lengthy, to make predictions.

## 3   Prompt Injection

In this section, we formally define Prompt Injection (PI) as a task and describe the benefits of the formulation. Prompt-dependent generation is a task of generating an output sequence $y$ that is a

proper response to the input sequence $x$ and coherent to the prompt $z$. Utilizing the prompt during inference, the generated sentence is obtained by $y = f(z, x)$ where $f$ denotes an LM such as T5 and GPT-2. Prompt Injection (PI), i.e., parameterization of prompts, allows LMs to perform prompt-dependent generation without using prompts during inference. To achieve this, we need to design a PI method $H$ to inject a prompt $z$ into an LM $f$. The process of PI can be represented as

$$f_z = H(z, f) \tag{1}$$

where $f_z$ denotes an LM injected with the prompt. Then the prompt-dependent output sequence can be obtained by $y = f_z(x)$.

PI can also be applied for long prompts whose length exceeds the LM's input sequence length. Given a long prompt $z$, we decompose it into multiple sub-prompts $\{z_i\}$ each of which fits the LM's input length, i.e., $z = z_{1:n} = [z_1; z_2; ...; z_n]$. Then the PI process can be executed iteratively, injecting each sub-prompt sequentially while the LM is aware of the previous sub-prompts:

$$f_{z_1} = H(z_1, f) \tag{2}$$
$$f_{z_{1:2}} = H(z_2, f_{z_1}) \tag{3}$$
$$\dots$$
$$f_{z_{1:n}} = H(z_n, f_{z_{1:n-1}}) \tag{4}$$

The above formulation can be seen as a high-level abstraction of iterative PI that we aim to approximate. In practice, in order to fully inject $z_{1:n}$, we repeat (2)-(4) multiple times (i.e., multiple epochs).

**Why is Prompt Injection necessary?** Prompt Injection brings definite advantages when applied to prompt-dependent tasks. The previous approach of appending prompts to the input sequences has the drawback of the model repeatedly referring to the prompt at each inference time. This becomes critical in scenarios requiring long prompts, as Transformer architecture has quadratic computational and memory costs due to the limitation of the self-attention operation. We propose PI as a solution to this computation bottleneck. Once a prompt is injected into the LM in advance, the LM no longer needs to refer to the prompt during inference. As a result, the model's input length remains independent of the length of prompts and is able to utilize prompts of any length efficiently. We discuss the efficiency gain of PI in Section 6.1.

**Evaluation Metric for Prompt Injection** PI can be evaluated by the evaluation metric of the fixed prompt-dependent task at hand. We also introduce a metric called the Prompt Injection score (PI score) to measure the degree of injection. The metric is agnostic of the target task by comparing the results with that of an LM given actual prompts during inference. Let $X_{w/\ prompt}$ denote the LM's task score with the prompt as an additional input (upper bound) and $X_{w/o\ prompt}$ denote the LM's task score without the prompt (lower bound). We define **PI score** as the min-max scaling score of $X_{PI}$, where $X_{PI}$ represents the score of the LM on the target task after PI, i.e., **PI score** $= \max(0, X_{PI} - X_{w/o\ prompt}) / (X_{w/\ prompt} - X_{w/o\ prompt})$. We limit using PI only in situations where $X_{w/\ prompt} > X_{w/o\ prompt}$ because there is no reason to inject a prompt if task performance degrades when using the prompt. Even if the range of individual task scores may vary from task to task, PI score represents the overall injection effectiveness of the PI methods, agnostic of the individual task score range.

# 4 Methods for Prompt Injection

In this section, we explore methods of Prompt Injection (PI) that can address prompt-dependent tasks without accessing the prompt during inference. To achieve this, the model should be trained to store the prompt in its parameters. This can be seen as *parameterizing* the prompt into the model instead of feeding the prompt explicitly to the model. This is challenging as the prompt is unseen to the model and has no corresponding training data. In Section 4.1, a baseline method by continued pre-training is introduced, followed by a method for improving the baseline with curriculum learning. Section 4.2 presents a novel distillation-based method called Pseudo-INput Generation (PING) that learns to generate pseudo-inputs to inject novel prompts.

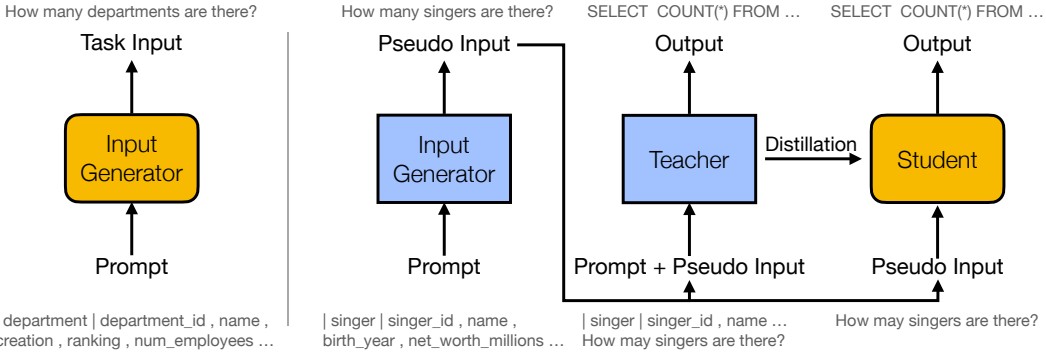

Figure 2: Illustration of the Pseudo-INput Generation (PING). During Phase 1, an input generator is trained with the task-specific training data. The inputs are prompts of a task, and the outputs are task inputs corresponding to the prompt. Input and output examples applied to semantic parsing are shown. During Phase 2, the input generator generates pseudo-inputs from the given target prompt, which are used to distill knowledge from the teacher to the student. Blue square boxes indicate frozen parameters; yellow rounded boxes indicate unfrozen parameters.

## 4.1 Continued Pre-training

We establish the Continued Pre-training method as a straightforward baseline for PI. This method injects prompts into the parameters of an LM by continuing with the pre-training objective of the LM on the target prompt. The pre-training objective is a straightforward option as it works in an unsupervised manner. In our experiments, we leverage the pre-trained T5 model [21] and thus use the masked language modeling objective which is the pre-training objective of T5. Following Raffel et al. [21], we randomly replace 15% of a given prompt with special mask tokens; then, the model is trained to predict the sequence of masked tokens. In this process, the model learns about the prompt the same way the model learns knowledge during the pre-training stage.

**Curriculum learning**    We further investigate the baseline method by leveraging *curricula* [2, 4] during continued pre-training. We set the mask ratio as the difficulty criteria [34] and gradually increase the ratio throughout the Continued Pre-training. As the mask ratio increases, the model should predict more masked tokens given less context. With curriculum learning, we expect the LM to gradually better adapt to the prompt, improving its prompt-dependent task performance. Throughout the experiments, we increase the mask ratio linearly from 15% to 30%, 50%, and 70% and report the best score.

## 4.2 Pseudo-INput Generation (PING)

The purpose of PI is to inject a prompt into the parameters of an LM which can also be done indirectly through distillation. In this subsection, we propose a novel distillation-based method called Pseudo-INput Generation (PING) that distills a novel prompt into a student LM that does not have access to the prompt through a teacher LM that does have access to the prompt. In order for distillation, pseudo-inputs are needed since we assume a scenario where the prompt to be injected has never been seen during training and does not have separate training data. An overview of PING is illustrated in Figure 2. As shown in the figure, during Phase 1, an input generator is trained with the task-specific training data. When given a prompt of the task as the input, the generator is expected to generate the task inputs that correspond to the prompt. During Phase 2, the input generator is frozen and is used to generate pseudo-inputs from the unseen prompt, which are then given to the teacher together with the prompt, while only the pseudo-inputs are given to the student. This way, the student learns to follow the teacher and is able to learn about the prompt indirectly. We believe that this is the first work that aims to distill knowledge with different inputs for the teacher and the student.

# 5 Experimental Setup

In this section, we explain the experimental setups in detail. All experiments are performed with the T5-base [21] (220M parameters) model unless noted otherwise.

## 5.1 Prompt-dependent tasks

In order to evaluate the effectiveness of Prompt Injection (PI) methods, we select three prompt-dependent tasks—persona-dependent conversation, semantic parsing, and zero-shot learning with task instructions; all these tasks require fixed prompts during inference. Fixed prompts come in the form of a persona in persona-dependent conversation [39], database schema in semantic parsing [10], and task instruction in zero-shot learning with task instructions [23]. As described in the introduction and Section 3, when PI is applied for these tasks, there would be apparent benefits in real world scenarios. For instance, PI eliminates the need to repeatedly include persona descriptions in the input during inference when serving a conversational model of a specific personality. With these tasks, not only the performance of the baseline PI methods is evaluated, but also the significance of PI is emphasized by comparison with the (unconstrained) previous approaches that concatenate prompts to the input.

## 5.2 Datasets

Following datasets of prompt-dependent tasks mentioned in Section 5.1 are utilized to evaluate Prompt Injection (PI).

**PERSONA-CHAT**    PERSONA-CHAT [39] is a crowd-sourced dataset intended for training agents to perform engaging and personal chit-chat by comprising the dialogues to be grounded on specific personas. They crowdsourced 1,155 unique personas, each with five profile sentences and 162,064 utterances over 10,907 dialogues. For each dialogue, two speakers have a 6-8 turn conversation conditioned on a given persona. The task is measured via perplexity (PPL). We randomly select 100 dialogues from the validation set as persona-dependent conversation benchmark for testing PI. The persona descriptions are 60 tokens long on average.

**Spider**    Spider [37] is a large cross-domain semantic parsing and text-to-SQL dataset for developing natural language interfaces to cross-domain databases. It includes 10,181 questions, 5,693 unique SQL queries, and 200 database schemas covering 138 different domains. Models must generalize to new database schemas as well as new queries to perform well on it. Evaluation metrics include Exact Matching (EM) and Execution Accuracy (EA). We utilize the dev set containing 20 databases with about 50 questions per database as a semantic parsing benchmark for PI. The database schemas range in length from 55 to 430 token lengths.

**WSC / RTE / COPA**    For the task of zero-shot task generalization, Raffel et al. [21] have trained the LM on a diverse set of tasks and evaluated on a held-out group of tasks to evaluate generalization performance. We choose coreference resolution, natural language inference, and sentence completion tasks, three out of their four held-out tasks, and test PI on WSC (Winograd Schema Challenge), RTE (Recognizing Textual Entailment), and COPA (Choice of Plausible Alternatives) datasets [30]. All of these tasks are binary classification tasks. We utilize task instructions (prompts) of WSC, RTE, and COPA provided from Raffel et al. [21] and report average task scores of using task instructions. The task instructions are comprised of about 20-30 tokens.

## 5.3 Implementation Details

For the Continued Pre-training method (Section 4.1), we use the Adam optimizer [15] with a constant learning rate 1e-4 and batch size 8. We perform 5-20 steps of injection. For PING (Section 4.2), input generators are trained on each tasks for 1-2 epochs. We use KL-divergence for distilling the last layer's output of the decoder and perform 10-40 steps of injection. Diverse pseudo-inputs are generated by sampling each token from the output probability distribution of the decoder. For all of the experiments except for zero-shot generalization, we use a single 16GB T4 GPU. For zero-shot generalization, we use 4 32GB V100 GPUs.

Table 1: Inference efficiency of different models that can be used for performing prompt-dependent inference. We depict how many times PI is efficient in comparison with the other approaches inside the parenthesis. When there is out-of-memory (OOM) using the 16GB T4 GPU, we estimate the FLOPs in *italics* assuming a linear correlation between prompt length and FLOPs.

| Model | Prompt Length | FLOPs (G) | Latency (s) |
|---|---|---|---|
| T5 w/ PI | * | 0.7k | 0.58 |
| T5 | 512 | 7.2k ($\times$10.3) | 1.09 ($\times$1.9) |
| | 512 $\times$ 2 | 14.6k ($\times$21.0) | 2.38 ($\times$4.1) |
| | 512 $\times$ 4 | OOM | - |
| T5 w/ FiD | 512 | 7.2k ($\times$10.3) | 1.09 ($\times$1.9) |
| | 512 $\times$ 2 | 14.0k ($\times$20.2) | 1.54 ($\times$2.6) |
| | 512 $\times$ 4 | 27.6k ($\times$39.8) | 2.87 ($\times$4.9) |
| | 512 $\times$ 8 | 54.9k ($\times$79.2) | 5.87 ($\times$10.0) |
| | 512 $\times$ 28 | OOM *($\times$280)* | - |
| Linear-Transformer | 512 | 9.5k ($\times$13.8) | 1.58 ($\times$2.7) |
| | 512 $\times$ 2 | 16.1k ($\times$23.2) | 2.62 ($\times$4.5) |
| | 512 $\times$ 4 | 29.2k ($\times$42.2) | 4.74 ($\times$8.1) |
| | 512 $\times$ 8 | 55.6k ($\times$80.1) | 9.11 ($\times$15.6) |
| | 512 $\times$ 28 | OOM *($\times$280)* | - |

In order for injection and comparison with upper-bound and lower-bound performance, we first need two different versions of the LM adapted to the given task. For the task of persona-dependent conversation and semantic parsing, one (upper bound) is fine-tuned together with prompts since prompts are explicitly used during inference, while the other (lower bound) is fine-tuned on the task without being given the prompt. We perform PI on the lower-bound LM since we also assume having no access to prompts during inference.

For the zero-shot learning task, we modify the prompts developed by Raffel et al. [21] in the form of a fixed prompt. Their prompts have placeholders such as `Premise`, and `Hypothesis`. We replace the placeholders with fixed words such as "Premise" and "Hypothesis", then append the actual content to the prompt in a key-value format. For example, if the original is `If {Premise} is true, is it also true that {Hypothesis}?`, then the converted prompt is `If "Premise" is true, is it also true that "Hypothesis"? Premise:{Premise} Hypothesis:{Hypothesis}`. This ensures that the prompt is fixed, which can be injected with PI. We use the T0-3B LM checkpoint for the zero-shot generalization.

## 6 Experimental Results

In this section, we first explore the inference efficiency of models performing prompt-dependent tasks and show that Prompt Injection (PI) leads to meaningful computational efficiency. Then the baseline and proposed methods are tested and compared on datasets discussed in Section 5.2. The results indicate that the Pseudo-INput Generation (PING) method achieves the best performance among PI methods, sometimes even outperforming the unconstrained upper bound, which uses explicit prompts during inference. In Section 6.3, we provide a concrete instance of injecting a real persona description into a conversational model, demonstrating the feasibility of long prompt injection.

### 6.1 Inference Efficiency

The comparison of inference efficiency of a model with PI, a baseline model that naively concatenates prompts to the input, Fusion-in-Decoder (FiD) [12], and Linear Transformer [13] are shown in Table 1. We consider FiD as one of the options for processing long inputs because it processes long input sequences by encoding chunks of input sequences separately, reducing the quadratic complexity to linear. Linear Transformer also reduces the complexity to linear by linearizing the

Table 2: Prompt Injection performance on three prompt-dependent tasks. W/ PROMPT stands for the upper bound (unconstrained) method, which uses the prompt during inference by appending it to the input. W/O PROMPT depicts the lower bound method of not utilizing the prompts at all. Lastly, we show three W/ PI methods: CP and CP W/ CURR stand for the Continued Pre-training (baseline) and the Continued Pre-training with curricular, respectively, as explained in Section 4.1; PING depicts our novel proposed method utilizing distillation.

| | Dialogue | | Semantic Parsing | | | Task Generalization | | | | | |
| | PERSONA-CHAT | | Spider | | | WSC | | RTE | | COPA | |
| | PPL (↓) | PI Score | EM | EA | PI Score | ACC | PI Score | ACC | PI Score | ACC | PI Score |
|---|---|---|---|---|---|---|---|---|---|---|---|
| W/ PROMPT | **8.83** | - | **57.9** | **61.3** | - | **63.6** | - | 67.9 | - | **67.3** | - |
| W/O PROMPT | | | | | | | | | | | |
|   W/O PI | 11.01 | - | 14.5 | 15.1 | - | 44.0 | - | 64.2 | - | 60.0 | - |
| W/ PI | | | | | | | | | | | |
|   CP | 10.85 | 0.073 | 16.9 | 17.5 | 0.054 | 54.5 | 0.536 | 67.7 | 0.946 | 64.8 | **0.658** |
|   CP W/ CURR | 10.61 | 0.183 | 17.7 | 18.4 | 0.072 | 50.8 | 0.347 | **68.2** | 1.08 | 64.1 | 0.562 |
|   PING | 9.82 | **0.546** | 36.6 | 41.7 | **0.507** | 63.7 | 1.005 | 64.2 | 0 | 60.6 | 0.082 |

attention mechanism. We measure FLOPs and forward propagation latency via DeepSpeed Flops profiler [2] using a single 16GB T4 GPU.

As shown in Table 1, T5 W/ PI is much more efficient than other models, especially as we assume a longer prompt length. This is because the efficiency of PI remains the same independent of the prompt length while the costs of others increase linearly. Specifically, when the prompt length is 8 times the model's max input sequence length, one can achieve $80\times$ computational efficiency in terms of FLOPs by applying PI. Furthermore, in a scenario where the prompt length is $28\times$ the model's max input sequence length (shown in Section 6.3 when trying to utilize a long persona that is over 13,000 token length long), previous approaches show an out-of-memory (OOM) issue using the 16GB T4 GPU, and it is impossible to utilize them. PI is estimated to be *280×* more efficient in terms of total FLOPs if there is no OOM issue.

## 6.2 Task Performance

In Table 2, we report the task performance obtained by applying different PI methods on three prompt-dependent tasks. PI scores are also obtained as introduced in Section 3. For all of W/ PI methods, we observe an overall increase in performance compared to W/O PROMPT, indicating successful injection of prompts into the parameters of the model through PI methods.

For the results, while CP gives modest performance improvement over W/O PROMPT, the results of CP W/ CURR show that leveraging curricula during continued pre-training is effective in some cases. CP W/ CURR performs better compared to CP in PERSONA-CHAT, Spider, and RTE; it even outperforms W/ PROMPT in RTE. On the other hand, PING significantly improves performance from CP in PERSONA-CHAT, Spider, and WSC, outperforming W/ PROMPT in WSC. This sheds light on the possibility that PI may be able to reach the upper bound (unconstrained) performance. However, the results show at the same time that there is still a gap between the performance of PI methods and the upper bound W/ PROMPT that needs to be bridged in future work.

We find that the performance of different methods depends on the complexity of the input sequence structure. We believe that PING achieves a good performance in PERSONA-CHAT, Spider, and WSC because those datasets have relatively simple input sequences (short utterances; simple query; a sentence and two words, respectively). In datasets with many components or multiple complex sentences (e.g., COPA and RTE), the low quality of generated pseudo-inputs degrades the performance of PING. On the other hand, CP and CP W/ CURR perform better in datasets with complex structure. These findings encourage the community to explore a more integral PI method that can cover different datasets.

---

[2]https://www.deepspeed.ai/tutorials/flops-profiler/

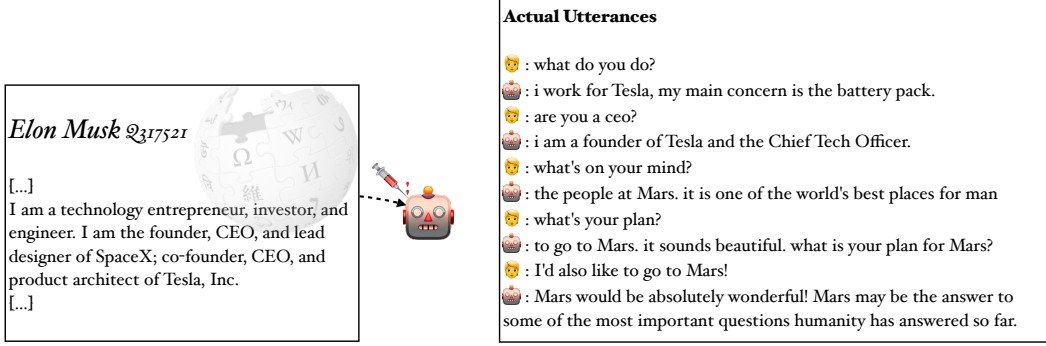

Figure 3: A real world example of Prompt Injection with a long prompt. (Left) The process of injecting a Wikipedia article describing a person (Elon Musk) into a model with PI. The article is more than 13,000 tokens long. (Right) Actual conversation between the persona injected model and a human that is hand-picked.

## 6.3 Long Prompts Injection

To demonstrate the effectiveness of PI on injection of long prompts into LMs, we show how the method works with a real world example. We pick a Wikipedia page (Elon Musk), considering it as a long persona description, and inject the entire article (over 13,000 tokens) into an LM trained with PERSONA-CHAT. Here, we use T5-large as a base model and apply PING.

Figure 3 shows an actual instance of interactions with the LM that underwent PI through PING. The responses show the LM successfully reflecting the description of the person on the Wikipedia page without having the description appended to the input. Moreover, the inference of PI is 280× more computationally efficient in terms of FLOPs than the baseline, as shown in Section 6.1. Lastly, we provide a live demo to allow interactions with an LM injected with the persona of Elon Musk.

## 7 Conclusion

**Limitations and Future Work**   While Prompt Injection (PI) enables performing prompt-dependent tasks efficiently, there are limitations that needs to be addressed in future work. In particular, the current PI methods cause task performance degradation. Moreover, the computational costs needed for the injection of prompts into the model parameters have not been extensively considered. For example, when considering *previous conversation history* as prompts to be injected in a multi-turn conversation setting, fast injection may also be a requirement for real-world application. Updating or adding a relatively small number of parameters [11, 31] may be a potential avenue for addressing the problems.

In this paper, we propose Prompt Injection (PI), a novel formulation of injecting the prompt into the parameters of an LM, as an efficient alternative to attaching fixed prompts to the inputs for prompt-dependent tasks. Through experiments, we show that PI is much more computationally efficient (up to 280 times) in terms of total FLOPs for handling long prompts compared to the previous alternatives. We further explore baseline methodologies for PI and find that Pseudo-INput Generation (PING), a distillation-based approach, shows promising results in persona-dependent conversation, semantic parsing, and zero-shot learning with task instructions. Through the explorations, we show that PI can be a promising direction for conditioning language models with prompts, especially in scenarios with long and fixed prompts. To this end, we hope the community explores PI for achieving both performance and efficiency on prompt-dependent tasks.

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
