# OpenReview forum: "Prompt Injection: Parameterization of Fixed Inputs"
_NeurIPS.cc/2022/Conference — NeurIPS 2022 Submitted_

### Official Review · Reviewer_y47p · 2022-06-24

**Rating:** 7
**Confidence:** 4
**Soundness:** 4 excellent
**Presentation:** 4 excellent
**Contribution:** 3 good

**Summary:**

This paper proposes a (self-proclaimed) novel approach called prompt injection, which injects the declarative knowledge of a prompt into the language model parameters by distilling a teacher model with the prompt into a student teacher without prompt on automatically generated inputs. The papers show that the injection is successful across a wide & diverse range of tasks and the approach saves inference time compute.

**Questions:**

- How many pseudo inputs did the paper distill on? I would imagine that the gap between the student and teacher should drastically decrease when you increase the number of pseudo inputs.
- How does the gap between student & teacher change w.r.t. model size? Should we expect future larger models to become better at this?
- Can you show some qualitative examples where you claimed that "the low quality of the pseudo generated inputs lead to lower performance"? How about the diversity of the pseudo inputs?

**Limitations:**

Yes.

**Strengths And Weaknesses:**

Strength:
- This paper elaborates on the impactful idea of "prompt-injection", and show that this is broadly useful for a wide range of tasks.
- This paper also sets up a reasonable evaluation framework that could be used to test future methods.
- This paper is generally well-written.

Weakness:
- The idea of prompt-injection is not novel, and various previous works have explored this idea. For example, [1] and "context distillation" in [2]. Please tune down the claim on novelty in the presentation. That said, I would still vote for acceptance of this paper since the other paper did not focus on the framing of prompt-injection, and this paper could be useful for the community to cite.
- I wished that there were more experiments and ablations, given that the contribution of this paper is more about "studying an existing algorithm (PING) and figure out its basic property", rather than "proposing a drastically novel algorithm".

[1] Towards Zero-Label Language Learning
[2] A General Language Assistant as a Laboratory for Alignment

---

> ### Author Response · Authors · 2022-08-02
> **Response to Reviewer y47p**
>
> Hello Reviewer y47p,
>
> Thank you for your valuable time on the extensive analysis highlighting the strengths of the paper!
>
> **Weaknesses**
> Weakness 1. Tone and presentation of the paper’s novelty: As the reviewer pointed out, we agree that prompt injection as a method is not entirely new, and our work rather focuses on formally framing the problem. Thank you for pointing out these missing works [1,2] and we will adjust the tone of our contributions and also properly credit them in the next revision.
>
> Weakness 2. Experiments and ablations: The authors agree that more experiments and ablations would enhance the paper. We have thus performed additional experiments with [Multi Session Chat](https://aclanthology.org/2022.acl-long.356.pdf) [3] dataset providing a more concrete example of where prompt injection is needed (long prompt & fast inference). Please refer to the general response for the details of the experimental results. The authors have also conducted ablation studies and presented the results while answering your questions below.
>
> **Questions**
> Question 1. Number of pseudo inputs: Since PING method creates new pseudo inputs every step for maximizing the diversity, we deemed that there is the concept of training steps but not the concept of the number of inputs.
>
> Question 2. Scale up models: We have increased the model size from t5-base (220M) to t5-large (770M) and trained the student and teacher model on PERSONA-CHAT dataset. It shows ppl 9.54 and 7.42 respectively, narrowing the gap between the student and teacher compared to the smaller model (t5-base) of ppl 11.01 and 8.83. The authors then performed prompt injection with PING method, showing the results of ppl 8.37 with PI Score 0.552, where the injection ability is better than the smaller model of PI Score 0.546. We will add the scale up experiments with larger models (t5-large, t5-3B) to the appendix.
>
> Question 3. Pseudo input quality and diversity: Here is an example of generated pseudo inputs showing the pseudo input quality of PERSONA-CHAT and RTE which leads to different injection performance (higher PI score in PERSONA-CHAT):
>
> PERSONA-CHAT: `<partner> that is good to know. do you have a favorite artist you can play in my shop?`
> RTE: `question: Which song splattered out in a jukebox? answer: "Assume your dream."`
>
> The generated input for PERSONA-CHAT is plausible while the generated input for RTE is not (RTE requires a premise and hypothesis). Moreover, in RTE, when we replaced the generated pseudo inputs of PING with the real inputs, we got results that even outperform w/ prompt (upper bound). It shows that higher quality of pseudo inputs will lead to higher prompt injection performance.
> Assuming 8 batch size and 100 distillation steps in PERSONA-CHAT, diversity was measured for 800 generated pseudo inputs, resulting in less than 20% duplicated pseudo inputs. All duplicated pseudo inputs are greetings that may not vary and it is aligned with the proportion of greetings in the partner’s utterances in the dataset.
>
> **Reference**
> [1] Zirui Wang and Adams Wei Yu and Orhan Firat and Yuan Cao. Towards Zero-Label Language Learning. ArXiv, abs/2109.09193, 2021.
> [2] Amanda Askell et al.. A General Language Assistant as a Laboratory for Alignment. ArXiv, abs/2112.00861, 2021.
> [3] Jing Xu and Arthur D. Szlam and Jason Weston. Beyond Goldfish Memory: Long-Term Open-Domain Conversation. In ACL, 2022.

---

> > ### Comment · Reviewer_y47p · 2022-08-03
> > **Thanks for the experiment. Decision to support unchanged.**
> >
> > Thanks for your response. My support for acceptance has not changed.
> >
> > Minor:
> >
> > - "Since PING method creates new pseudo inputs every step for maximizing the diversity, we deemed that there is the concept of training steps but not the concept of the number of inputs."
> >
> > I might have missed it in the paper -- how does the number of injection steps influence the performance? Intuitively, if you train for infinite steps and cover the entire space of samples, the teacher-student gap should decrease, is that right?

---

> > > ### Author Response · Authors · 2022-08-04
> > > **Response to Reviewer y47p**
> > >
> > > Hello Reviewer y47p,
> > >
> > > Thank you for the question.
> > >
> > > Yes, intuitively, the teacher-student gap should decrease if the samples cover a higher portion of the input space. The performance usually converges at 100-200 steps and is saturated thereafter.

---

> > > > ### Comment · Reviewer_y47p · 2022-08-04
> > > > **Response**
> > > >
> > > > Thanks! Just curious, what would happen if you increase the temperature when sampling the pseudo input? That might increase the diversity of the outputs.
> > > >
> > > > (Don't feel pressured to respond or run additional experiments =P)

---

> > > > > ### Author Response · Authors · 2022-08-04
> > > > > **Response to Reviewer y47p**
> > > > >
> > > > > Hello Reviewer y47p,
> > > > >
> > > > > Thank you for your interest in our work.
> > > > >
> > > > > We increased the temperature from 1 to 2 in preliminary experiments, and the performance decreased. We deemed that it was because the quality of generated inputs decreased due to the generation of sentences that don’t match the syntax and make no sense. Therefore we set the temperature to 1 throughout the experiments.

---

### Official Review · Reviewer_XfKG · 2022-07-02

**Rating:** 5
**Confidence:** 3
**Soundness:** 3 good
**Presentation:** 3 good
**Contribution:** 2 fair

**Summary:**

The paper is about prompt injection, which aims to be an efficient alternative to attaching fixed prompts to the input. Prompt injection is sample efficient and worth it when there are long prompts. This can be useful if there is a detailed description of a personalized chatbot, for example.

**Update after response.**

With the additional experiment, I've increased my score to 5.

**Questions:**

How does the proposed method compare with other baselines such as context distillation? (Askell et al, https://arxiv.org/pdf/2112.00861.pdf)
Are there other approaches where this approach might be especially useful? As the main benefit is inference-time efficiency, one might imagine that the WSC, RTE, COPA, and SPIDER tasks are not the prime candidates for this approach. The dialogue eval is good but it only uses one dataset.

**Limitations:**

Seems fine to me

**Strengths And Weaknesses:**

Originality: Paper tackles the problem of using fixed prompts many times for different examples.
Quality: Experiments done in the paper are OK but not particularly stunning. One way to improve the experiments could have been to pick a setting where it was crucial for long contexts (like summarization, or long dialogues), and show that with this method it was possible to use longer contexts which lead to substantially better performance. Or show that it works over a broad range of settings.
Clarity: The paper is relatively clear.
Significance: The paper proposes a method for prepending fixed prompts in a computationally efficient manner.

While the paper is a nice start, I think it still leaves a lot to be desired. There is enough substance to warrant an ACL short paper, but I am not sure about a NeurIPS paper. Some examples of how I think the paper could be stronger:
- Empirical results showing that PING works across a very wide range of tasks with good performance.
- Comparison with other methods that try to do similar things.
- Show for tasks that require long inputs and long prompts, PING enables substantial performance improvements.
- Overall, experiments on settings where fast inference is crucial might make it more clear to the reader why it is important to trade off this complexity from using PING for faster inference.

Minor note: Line 22 should be Sanh et al, not Raffel et al., right?
Some further related work might have been missed, such as context distillation (https://arxiv.org/pdf/2112.00861.pdf)

---

> ### Author Response · Authors · 2022-08-02
> **Response to Reviewer XfKG**
>
> Hello Reviewer XfKG,
>
> Thank you for your valuable time in reviewing our paper and providing valuable comments to improve our work. The authors agree that performing additional experiments that you have mentioned would enhance the motivation of Prompt Injection.
>
> **Weaknesses**
> Weakness 1. More experiments: As you suggested, we have additionally considered a long dialogues setting where it is crucial to utilize long contexts as well as require fast inference to make proper and engaging responses. Please see the general response for the details of the experiments. In short, we have observed that Prompt Injection enables fast inference with long prompts, achieving moderate performance as a result of the tradeoff, but it is expected to be improved with enhanced injection methods.
>
> **Questions**
> Question 1. Comparison with baselines: context distillation [2], which can be considered as a special case of Prompt Injection, is not directly applicable to our setup as it requires training data for distillation. We will include more discussion about such baselines in the future revision.
>
> Question 2. Settings that benefit from Prompt Injection: The authors want to say that Prompt Injection approach is especially useful for dialogue settings. As the benefits are inference efficiency and the use of long prompts, dialogue tasks that require real-time response and have long contexts such as long persona description or long dialogue history, can benefit from Prompt Injection approach. Also, we can imagine having very long database schemas for semantic parsing tasks for real-world application. The authors agree that task generalization (WSC, RTE, and COPA) is not the prime candidate for this approach, but wanted to show that Prompt Injection can be extended to any settings where prompts are used. We have thus performed additional experiments with MSC [1] dataset providing a more concrete example of where prompt injection is needed (long prompt & fast inference).
>
> **Reference**
> [1] Jing Xu and Arthur D. Szlam and Jason Weston. Beyond Goldfish Memory: Long-Term Open-Domain Conversation. In ACL, 2022.
> [2] Amanda Askell et al.. A General Language Assistant as a Laboratory for Alignment. ArXiv, abs/2112.00861, 2021.

---

### Official Review · Reviewer_AvLV · 2022-07-12

**Rating:** 7
**Confidence:** 3
**Soundness:** 3 good
**Presentation:** 4 excellent
**Contribution:** 3 good

**Summary:**

This paper addresses the problem in prompt learning that prompts are always included in the input text, which can cause computational and memory overhead.
To this end, this paper proposes prompt injection, a method that injects the prompt into the parameters of a pre-trained language model (PLM).
They show that the prompt injection method can be 280 times more efficient compared with previous approaches.

**Questions:**

NA

**Strengths And Weaknesses:**

Strength:
1. This paper proposes an interesting question that many conditional text generation tasks usually include an instance-specific prompt for each data, for example the persona profile for persona dialogue generation.
However, such prompt can cause computational costs and make it difficult for models to capture the key information during inference, in particular when the prompt is very long.
Therefore, this paper proposes a new method call Prompt Inject (PI) to store the prompts in the PLM instead of concating it to the input.
To my knowledge, the idea is very novel and interesting.

2. The experiments are solid.
This paper conducts experiments on persona dialogue generation, Text-2-SQL and zero-shot task generalization.
The experiments show that the PI can speed up the inference time and improve the lower bound of the models.

3. This paper is well written with proper citation and clarification

Weakness:
A minor note would be that the prompt in this paper is different from the prompt in most sense in other NLP paper, such prefix tuning, prompt tuning, etc.
So I  would suggest that the authors could give a more clear explanation on this issue in the introduction.

---

> ### Author Response · Authors · 2022-08-02
> **Response to Reviewer AvLV**
>
> Hello Reviewer AvLV,
>
> Thank you for providing us with an analysis and positive feedback! Also, the authors are very glad to see your interest in our work.
>
> > Weakness: A minor note would be that the prompt in this paper is different from the prompt in most sense in other NLP paper, such prefix tuning, prompt tuning, etc. So I would suggest that the authors could give a more clear explanation on this issue in the introduction.
>
> We will add clarification about what we refer to as “prompt” in the introduction as below:
>
> *Note that the "prompt" in this work refers to a broader aspect of prompts which includes both the prompts used for large language models to induce specific behavior as well as prompts used for smaller language models to provide some contextual knowledge such as persona for dialogue agents or database schema for semantic parsing models.*

---

> > ### Comment · Reviewer_AvLV · 2022-08-03
> > **Thank you for the clarification.**
> >
> > Thank the authors for their clarification on the term ''prompt''.
> > Overall, I think this is a good paper that can be accepted.

---

### Author Response · Authors · 2022-08-02
**General Response**

We thank all reviewers for their valuable comments. We have individually responded to each reviewer’s comment, and we put the details of the additional experimental results in this general response.

As both Reviewer XfKG and y47p suggested, we have additionally considered a long-dialogue setting. We have picked [Multi Session Chat](https://aclanthology.org/2022.acl-long.356.pdf) (MSC) [1] dataset that consists of multiple chat sessions intended to establish long-term conversations. We have considered both persona and conversation history as prompts and applied Prompt Injection to the MSC dataset. The experiment results are as follows:
|               | PPL               | PI Score |
| :--------     | :----:             | :----: |
| w/ Prompt     | **13.8**          |       |
| w/o Prompt    | 23.92             |       |
| w/ PI         |                   |       |
| CP            | 18.55             | 0.531 |
| CP w/ curr    | 18.37             | *0.549* |
| PING          | *17.96*           | **0.590** |

The results show the same trend as in PERSONA-CHAT with regard to injection performance, achieving even better PI Scores  0.531, 0.549, and 0.589 compared to the PI Scores of PERSONA-CHAT which are 0.073, 0.183, and 0.546. It demonstrates that Prompt Injection enables fast inference with long prompts, achieving moderate performance as a result of the tradeoff, but it is expected to be improved with enhanced injection methods. This demonstrates the usefulness of Prompt Injection in real-world settings that require long prompts and fast inference.

**Reference**
[1] Jing Xu and Arthur D. Szlam and Jason Weston. Beyond Goldfish Memory: Long-Term Open-Domain Conversation. In ACL, 2022.

---

### Meta-Review · Area_Chair_zBsx · 2022-08-29

**Recommendation:** Reject
**Confidence:** Less certain

**Metareview:**

The paper proposes a method for distilling prompts into the parameters of a model. Reviewers liked that the method can improve the efficiency of inference by avoiding having to attend over prompts, and the evaluation on the PersonaChat dataset is a good use case for this approach. However, several important concerns were raised. As pointed out by reviewer y47p, similar ideas have been explored in previous work, and claims of novelty need to be toned down. As acknowledged in the author response, most of the experiments are on tasks with short inputs, so gain little benefit from the approach. The difficulty in finding suitable tasks where the approach has a clear benefit might suggest the method has limited applicability. The additional experiment on MSC is a nice addition in the author response, although results here are a bit underwhelming. Overall, this is borderline, leaning reject.



**Award:**

No

---

### Decision · Program_Chairs · 2022-09-14

Reject